# Group Velocity Modulation and Light Field Focusing of the Edge States in Chirped Valley Graphene Plasmonic Metamaterials

**DOI:** 10.3390/nano11071808

**Published:** 2021-07-12

**Authors:** Liqiang Zhuo, Huiru He, Ruimin Huang, Shaojian Su, Zhili Lin, Weibin Qiu, Beiju Huang, Qiang Kan

**Affiliations:** 1College of Information Science and Engineering, Huaqiao University, Xiamen 361021, China; lqzhuo@stu.hqu.edu.cn (L.Z.); eeclab@hqu.edu.cn (H.H.); sushaojian@hqu.edu.cn (S.S.); zllin@hqu.edu.cn (Z.L.); 2Institute of Semiconductors, Chinese Academy of Sciences, Beijing 100086, China; bjhuang@semi.ac.cn (B.H.); kanqiang@semi.ac.cn (Q.K.)

**Keywords:** valley degree of freedom, edge states, chirped valley graphene plasmonic metamaterial waveguide, excellent tunability, group velocity modulation

## Abstract

The valley degree of freedom, like the spin degree of freedom in spintronics, is regarded as a new information carrier, promoting the emerging valley photonics. Although there exist topologically protected valley edge states which are immune to optical backscattering caused by defects and sharp edges at the inverse valley Hall phase interfaces composed of ordinary optical dielectric materials, the dispersion and the frequency range of the edge states cannot be tuned once the geometrical parameters of the materials are determined. In this paper, we propose a chirped valley graphene plasmonic metamaterial waveguide composed of the valley graphene plasmonic metamaterials (VGPMs) with regularly varying chemical potentials while keeping the geometrical parameters constant. Due to the excellent tunability of graphene, the proposed waveguide supports group velocity modulation and zero group velocity of the edge states, where the light field of different frequencies focuses at different specific locations. The proposed structures may find significant applications in the fields of slow light, micro–nano-optics, topological plasmonics, and on-chip light manipulation.

## 1. Introduction

Valley degrees of freedom (DOF), also known as valley pseudospins, mark discrete extreme energy states in the momentum space [1,2,3,4,5]. Valley pseudospin appears widely not only in conventional semiconductor materials, but also in classical wave artificial crystals, such as phononic crystals [6,7,8,9,10,11,12,13] and photonic crystals (PhCs) [14,15,16,17,18,19,20,21,22,23,24,25,26,27]. Similar to the spin DOF in spintronics, the valley DOF is regarded as a new information carrier [18,23,24] and provides a more effective method of dealing with the nontrivial topological phase [21,22,23,24,25] which makes valley topological photonics become a research field in the current frontier. In order to obtain the valley topological phase, Berry curvature at the K and K’ valley in the Brillouin zone (BZ) is obtained by breaking the spatial inversion symmetry of photonic crystals [18,19,20,21,23,24,25] in which the scatterers are arranged in a periodic honeycomb. On the other hand, the eigenmode of the K and K’ valley in the BZ of such a structure have opposite chiral orbital angular momentum (OAM) [18]. Thus, topological transmission is realized. Moreover, the original Dirac cones at the K and K’ points are opened to form a complete photonic bandgap, resulting in the edge states at the interfaces of different valley Hall phases [23,24,25], which are immune to backscattering [28,29] caused by defects and acute light channels. In recent years, valley topological insulators (TIs) have been widely studied in optical, acoustic, and electronic systems, such as valley topological robust transport [18,19,20], topological photon routing [24,30], unidirectional light transport [24], valley topological edge state frequency tuning [13], valley topological acoustic wave group velocity modulation based on phononic crystals [13], and topological spin–valley filtering effects [31,32]. These studies have opened up unprecedented application opportunities for valley TIs in the fields of tunable acoustics, topological photonics, and the emerging field of nontrivial states. Although topological materials with valley DOF greatly promote and optimize the transmission of information and energy, they still face significant challenges in practical application due to their poor tunability and weak anti-scattering ability.

Graphene, which exhibits excellent properties, has promising applications in various fields [33,34,35,36]. More specifically, graphene plasmonic metamaterials (GPMs) [37,38,39], a kind of graphene-based metamaterials, have attracted extensive attention and research due to their unique Dirac conical band structure, compact field constraints [40,41], relatively low propagation loss [41,42], and flexible tunability [43,44,45,46]. Although traditional optical topological materials are still widely used, their limitations are apparent. Once the structural parameters of the valley photon topological insulators composed of traditional optical topological materials are determined, the operating frequency range cannot be changed, and tunability is extremely limited. Therefore, compared with traditional topological materials, GPMs have advantages. Moreover, since the operating frequency range of GPMs from near-infrared to terahertz is electrically and chemically adjustable, the plasmonic devices based on GPMs have gained wider attention. In recent years, thanks to the efforts of Xiong et al., GPMs have been realized and have great application potential in the fields of integrated, micro–nano-optics, and on-chip light manipulation [39].

In this paper, we propose tunable valley GPMs (VGPMs) for group velocity modulation and light field focusing of surface plasmon polariton (SPPs) waves. It consists of graphene nanodisks arranged in a honeycomb lattice covered with a graphene monolayer on the top. Tunable VGPMs take the valley as the DOF and break the spatial inversion symmetry by changing the chemical potentials, thereby opening a complete photon bandgap in the entire BZ. The nontrivial topology phase transition is confirmed by verifying the valley Chern numbers. Furthermore, we designed a chirped VGPM waveguide composed of a supercell arrangement of VGPMs with gradually increasing chemical potential difference. By implementing the chirped VGPM waveguide, we demonstrated modulation of the edge state group velocity. Group velocity was slowed down to zero to realize the slow SPP wave and the light field of SPP wave focusing. The chirped VGPM waveguide has excellent potential in nanophotonic systems, alternative topological states, and the manipulation of spin–orbit interactions of light because of its excellent tunability, backscattering resistance, and low absorption.

## 2. Calculation Methods and Models

As depicted in Figure 1, the designed VGPMs consisted of monolayer graphene, a silica layer, and a silicon substrate with the cylinder of periodic thicknesses distributed in 2D honeycomb lattices. Graphene regions with silica heights $h_{1}$, $h_{2}$, and $h_{3}$ have different chemical potentials $\mu_{c1}$, $\mu_{c2}$, and $\mu_{c3}$ under a back-gate bias voltage. The chemical potentials ratio $\mu_{c1}:\mu_{c2}:\mu_{c3}$ between the three graphene regions with different silica heights under an external gate voltage is equal to ${\left(h_{3}:h_{2}:h_{1}\right)}^{1/2}$ [47]. Therefore, chemical potentials of VGPMs can be modulated periodically by changing the heights of the silica layer and the back-gate bias voltage. In our design, $h_{1}$ was fixed at 140 nm and the chemical potential ratios were regulated by $h_{2}$ and $h_{3}$. When $\mu_{c1}=\mu_{c2}$, the Dirac cone dispersion at the K and K’ points in the first BZ of VGPMs was protected by both spatial inversion symmetry and time reversal symmetry. In this case, the band structure of VGPMs did not have a complete bandgap. By changing the chemical potential of the two graphene nanodisks so that $\mu_{c1}>\mu_{c2}$ (or $\mu_{c1}<\mu_{c2}$), VGPM1 (or VGPM2) with broken spatial inversion symmetry could be obtained. Here, because VGPM2 was the antisymmetric partner of VGPM1, VGPM1 and VGPM2 had the same band structure. Remarkably, in general photonic crystals, it is usually necessary to construct a unit cell with two nonequivalent dielectric cylinders to break the spatial inversion symmetry. This results in no further tuning in the process once the crystal structures are determined. In contrast, VGPMs have the overwhelming advantage of tuning without changing the lattice geometry. It leads to VGPMs with more flexible tunability and broader applicability. COMSOL Multi-Physics, which is a commercial finite element method (FEM) software suite, was used to calculate the band structures, transmission, and light field focusing of VGPMs in this study.

The significance of the graphene chemical potential is wholly demonstrated in the dispersion relation of the transverse magnetic (TM) polarized SPP mode supported on the monolayer of graphene. This dispersion relation is derived by solving Maxwell’s equations with boundary conditions, which is described in [48] as
(1)$$\frac{\varepsilon_{Air}}{\sqrt{\beta^{2}-k_{0}^{2}\varepsilon_{Air}}}+\frac{\varepsilon_{Silica}}{\sqrt{\beta^{2}-k_{0}^{2}\varepsilon_{Silica}}}=\frac{\sigma_{g}}{i\omega \varepsilon_{0}},$$
where $\varepsilon_{Air}$ and $\varepsilon_{Silica}$ represent the relative permittivity of air and silica corresponding to the upper region and the substrate; ω and $\varepsilon_{0}$ are the angular frequency of the plasmon and the vacuum permittivity of free space, respectively; $k_{0}=2\pi /\lambda$ stands for the vacuum wave number with the operating wavelength λ in the vacuum. In the nonretarded regime [48], the vacuum wave number $k_{0}$ is much smaller than the propagation constant of SPPs β, i.e., $k_{0}\ll \beta$. Thus, Equation (1) is simplified to
(2)$$\beta =\varepsilon_{0}\frac{\varepsilon_{Air}+\varepsilon_{Silica}}{2}\frac{2i\omega }{\sigma_{g}},$$

Here, $\sigma_{g}$, depicted as complex-valued surface conductivity of graphene, is composed of the interband electron transitions $\sigma_{inter}$ and the intraband electron–photon scattering $\sigma_{intra}$; see the Kubo formula [49]:(3)$$\sigma_{g}=\sigma_{intra}+\sigma_{inter},$$
with
(4)$$\sigma_{intra}=\frac{ie^{2}k_{B}T}{\pi h^{2}\left(\omega +i/\tau \right)}\left\{\frac{\mu_{c}}{k_{B}T}+2ln\left[1+\exp\left(-\frac{\mu_{c}}{k_{B}T}\right)\right]\right\},$$
(5)$$\sigma_{inter}=\frac{ie^{2}}{4\pi \hbar }ln\left[\frac{2\left|\mu_{c}\right|-\hbar \left(\omega +i/\tau \right)}{2\left|\mu_{c}\right|+\hbar \left(\omega +i/\tau \right)}\right].$$

The constants e, $k_{B}$, and ℏ denote the electron charge, the Boltzmann constant, and the reduced Planck constant, respectively; $\sigma_{intra}$ and $\sigma_{inter}$ are governed by the temperature T, the chemical potential $\mu_{c}$, the angular frequency of the plasmon ω, and the electron momentum relaxation time τ. Under the reasonable conditions of setting various parameters, the complex refractive index of SPP modes on the graphene layer is expressed as $n_{eff}=\beta /k_{0}$. It is worth noting that, in a specific SPP mode, the refractive index $n_{eff}$ depends only on the chemical potential $\mu_{c}$. Thus, graphene monolayers with specific periodicity $\mu_{c}$ act as VGPMs.

## 3. Results and Discussion

### 3.1. Topological Phase Transition of VGPMs

Figure 2a is a schematic diagram of the arrangement of VGPMs graphene nanodisks. The red dotted lines are the unit cells of the lattice. Each unit cell contained two non-equivalent graphene nanodisks with the same radius r of 0.21a, where the lattice constant a=40 nm. The chemical potentials of the two graphene nanodisks and the ambient graphene were $\mu_{c1}$, $\mu_{c2}$, and $\mu_{c3}$, respectively. We set the chemical potential of the ambient graphene $\mu_{c3}$ to 0.6 eV and kept it constant. When $\mu_{c1}=\mu_{c2}=0.3\;\mathrm{eV}$, the photonic band structure of VGPMs in the BZ featured Dirac degeneracies at the K and K’ points (blue dotted line in Figure 2b). It is worth noting that the group symmetry of the honeycomb lattice produces Dirac degeneracies. In this case, the Dirac degeneracies at the K and K’ valley were protected by the spatial inversion symmetry and the time reversal symmetry. When one of the symmetries is broken, the Dirac band structure immediately degenerates, and a complete photonic bandgap is opened. By modulating the chemical potentials of the two graphene nanodisks, the VGPM1 with $\mu_{c1}>\mu_{c2}$ or VGPM2 with $\mu_{c1}<\mu_{c2}$ were obtained. Here, the chemical potential difference $\Delta \mu_{c}$ was defined as $\mu_{c1}-\mu_{c2}$ to describe the amplitude of inversion symmetry breaking. When $\Delta \mu_{c}=\mu_{c1}-\mu_{c2}=0.08\;\mathrm{eV}$, the inversion symmetry was broken, resulting in the elimination of the Dirac degeneracies and the appearance of a complete photonic bandgap (solid red line in Figure 2b). The insets in Figure 2b demonstrate the unit cell of the VGPMs (left) and the first BZ with high symmetry points (right).

Figure 2c illustrates the topological index distributions for VGPMs with $\Delta \mu_{c}<0\;\mathrm{eV}$ (VGPM1) and $\Delta \mu_{c}>0\;\mathrm{eV}$ (VGPM2). The nonzero valley Chern number C_v_ = C_K_ − C_K’_ was used to distinguish the topology, where C_K_**/** C_K’_ is the valley-dependent index at the K/K’ valley. For VGPM1 with $\Delta \mu_{c}<0\;\mathrm{eV}$, the valley Chern number C_v_ = −1; for VGPM2 with $\Delta \mu_{c}>0\;\mathrm{eV}$, C_v_ = 1. These characteristics theoretically reveal the valley Hall phase transition. Furthermore, this is also verified in the numerical simulation corresponding to the topological index of the left and right sides in Figure 2c. Figure 2d presents the evolution of the band-edge frequencies at the K_1_ and K_2_ valleys (marked in Figure 2b) versus the chemical potential difference $\Delta \mu_{c}$ while keeping the average chemical potential $\left(\mu_{c1}+\mu_{c2}\right)/2$ unchanged. Note that the photonic bandgap width boosted continuously as $\left|\Delta \mu_{c}\right|$ increased from zero. The photonic light fields and power flux fields for the K_1_ and K_2_ valleys are shown in Figure 2e, where $\Delta \mu_{c}=0.08\;\mathrm{eV}\;\left(\mathrm{or}\;–0.08\;\mathrm{eV}\right)$. The light fields and power flux fields at the K_1_ and K_2_ valleys showed different vortex directions, respectively. When $\Delta \mu_{c}<0\;\mathrm{eV}$, the power flux fields at the K_1_ and K_2_ valleys presented clockwise (blue arrow) and anticlockwise (green arrow) power flows, i.e., pseudospins, around the field centers. This case was the opposite when $\Delta \mu_{c}>0\;\mathrm{eV}$. The frequencies of the four states in Figure 2e are marked in Figure 2d. At this point, it is intuitively seen in Figure 2e that the photonic light fields and power flux fields at the K_1_ and K_2_ valleys were reversed with the increase of $\Delta \mu_{c}$. This indicates that the energy bands of K_1_ and K_2_ were reversed and accompanied by a valley Hall phase transition.

### 3.2. Valley Topological Edge States of VGPMs

The valley topological edge state is a vital feature of a VGPM and an essential foundation for the designed chirped VGPM edge state waveguide. The valley topological interface used to realize the valley topological edge state is shown in Figure 3a. It is constructed by two inversion-symmetry broken VGPMs, i.e., VGPM1 and VGPM2. Here, the chemical potential difference $\Delta \mu_{c}=\mu_{c1}-\mu_{c2}$ of VGPM1 was −0.08 eV, where $\mu_{c1}=0.26\;\mathrm{eV}$ and $\mu_{c2}=0.34\;\mathrm{eV}$. To make sure the average chemical potential $\left(\mu_{c1}+\mu_{c2}\right)/2$ was consistent, we set $\mu_{c1}=0.34\;\mathrm{eV}$, $\mu_{c2}=0.26\;\mathrm{eV}$, and $\Delta \mu_{c}=\mu_{c1}-\mu_{c2}=0.08\;\mathrm{eV}$ for VGPM2. The red dotted rectangle in Figure 3a is a supercell for the valley topological interface. Figure 3b illustrates the valley topological edge states (magenta line) for the supercell. The Dirac degeneracy at the K/K’ point disappeared, and a complete photonic bandgap was formed, in which valley topological protective edge states existed. The grey-shaded region represents the projected bulk band for the TM polarized band. The insets in Figure 3b show the valley topological edge state at the K and K’ points. Color and white arrows indicate the photonic light field and the power flux, respectively. It is clearly observed that the light field flow vortices on both sides of the valley topological interface were obviously opposite. In addition, Figure 3c reveals the photonic light field of the supercell at the K/K’ point, indicating that the largest part light field was concentrated near the interface, which confirms that this was an edge mode.

### 3.3. Group Velocity Modification and Light Field Focusing of the Valley Topological Edge States in Chirped VGPMs

The dispersion of the valley topological edge states is modulated by varying $\Delta \mu_{c}$. As depicted in Figure 4a, the frequency at the highest point of the valley topological edge state gradually decreased with the increase of $\Delta \mu_{c}$. Combined with the group velocity equation $v_{g}=d\omega /dk$, it was calculated that the group velocity at the highest point of the valley topological edge state was zero. By appropriately modulating the chemical potentials of the valley topological interface, group velocity modulation and light field focusing of SPPs with topological protection were achieved. Therefore, a schematic diagram of the chirped VGPM waveguide modulated by the valley topological interface is shown in Figure 4b. The red and blue circles represent the locations of the weak and the strong chemical potentials. The nanodisks of weak chemical potential and strong chemical potential in the *i*-column are denoted as $\mu_{c-wk,i}$ and $\mu_{c-st,i}$, respectively. As i increases linearly, they following is true: $\mu_{c-wk,i}=\mu_{c-wk,1}-\left(i-1\right)\mu_{c-\delta }$ and $\mu_{c-st,i}=\mu_{c-st,1}+\left(i-1\right)\mu_{c-\delta }$, where $\mu_{c-\delta }$ is the step size and $\mu_{c-st,i}-\mu_{c-wk,i}=\mu_{c-st,1}-\mu_{c-wk,1}+2\left(i-1\right)\mu_{c-\delta }$ is the chemical potential difference $\Delta \mu_{c}$. In our case, we set $\mu_{c-wk,1}=0.27\;\mathrm{eV}$, $\mu_{c-st,1}=0.33\;\mathrm{eV}$, and $\mu_{c-\delta }=0.005\;\mathrm{eV}$. Figure 4c plots the dispersion relations of supercells in columns i = 1 and 31. It is intuitively observed that the frequencies of the edge state (red and blue line) decrease as m boosts. In addition, we were also interested in the group velocity $v_{g}$ dispersion curves of the valley topological edge states. Figure 4d shows the group velocity $v_{g}$ dispersion curves for different chemical potential differences $\Delta \mu_{c}$ from 0.07 eV to 0.14 eV. By comparing these curves, it was found that the larger the chemical potential difference $\Delta \mu_{c}$, the smaller the group velocity $v_{g}$ of the same frequency; furthermore, the frequency with zero group velocity was redshifted. Therefore, when the chirped VGPM waveguide is excited at a frequency where the group velocity of the edge state with a specific chemical potential difference is zero, the group velocity and light field of the SPPs can be modulated at the interface location of this chemical potential difference.

The group velocity dispersion curves and the light field distributions of edge SPP waves at different frequencies along the chirped VGPM waveguide interface are shown in Figure 5. Figure 5a–c plots the calculated group velocities $v_{g}$ versus the chirped VGPM waveguide location x at different frequencies of 50.30, 50.23, and 50.14 THz, respectively. The mutual feature of Figure 5a–c is that the group velocities gradually slowed down along the +x direction and finally decreased to zero. By comparing the three figures, it is evident that the x location of the zero group velocity increased as the frequency decreased, which means the edge SPP waves could transmit a longer distance along the +x direction at a lower frequency. Therefore, the lower the frequency, the farther away the light field accumulates. The light field distributions of the edge SPP waves are depicted in Figure 5d–f corresponding to Figure 5a–c, respectively. The highlighted areas of the photonic light field distributions are the locations where the edge SPP waves’ light field accumulated. It is seen more intuitively from the photonic light field distributions that the edge SPP waves of different frequencies are confined to the specific positions of the chirped VGPM waveguide, and the lower the frequency, the more restricted position is offset to the +x direction. Besides, the normalized light field distributions of the edge SPP waves at the interface of the chirped VGPM edge state waveguide extracted from the light fields (Figure 5d–f) are shown in Figure 5g–i. The normalized light field distributions indicate that the edge SPP waves at different frequencies stopped propagating forward at different x locations. By longitudinal comparison, the locations at which the group velocity was zero (Figure 5a–c) were well-matched with the locations at which the numerically experimental edge SPP wave stopped propagating (Figure 5d–i). Thus, the group velocity curve could predict where a wave of a specific frequency would stop propagating. The reliability of slow edge SPP waves was further verified by introducing the intensity enhancement factor $ℛ=|E_{max}{|}^{2}/|E_{0}{|}^{2}$, where $|E_{max}{|}^{2}$ is the maximum light field intensity of the edge SPP wave and $|E_{0}{|}^{2}$ is the light field intensity of the excitation source. The intensity enhancement factor R for the normalized light field distributions with different frequencies of 50.30, 50.23, 50.14 THz reached 34.45, 1.09, and 1.21, respectively.

## 4. Conclusions

In summary, we designed VGPMs, verified the valley pseudospin and valley hall phase transition, and numerically calculated the band structure. The valley topological interface was constructed by using the reverse valley Hall phases, and the dispersion curve of the valley topological edge mode was obtained. Then, the dispersion of the valley topological edge mode was tuned by changing the chemical potential difference of VGPMs, and the chirped VGPM waveguide was proposed accordingly. This waveguide offers multiple functionalities, including tuning of dispersion relations for valley topological edge states, modulating the group velocity of edge SPP waves, and realizing slow SPP waves. Finally, the intensity enhancement factor is introduced to verify the reliability of slow SPP waves and the light field focusing of the edge SPP waves. The proposed structure with excellent properties might find broad application in the fields of nanophotonic systems, alternative topological states, and manipulation of spin–orbit interactions of light.

## Figures and Tables

**Figure 1 nanomaterials-11-01808-f001:**
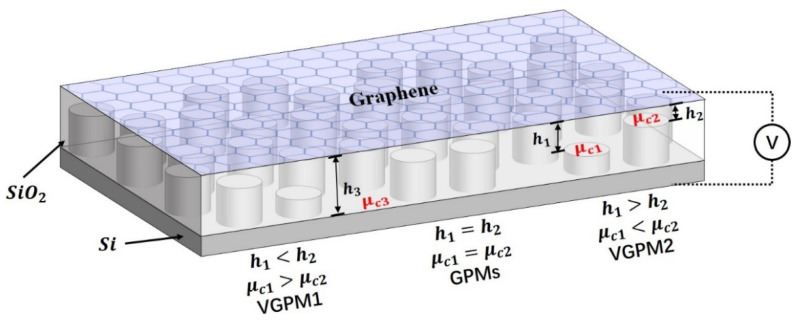
Schematic of valley graphene plasmonic metamaterials (VGPMs).

**Figure 2 nanomaterials-11-01808-f002:**
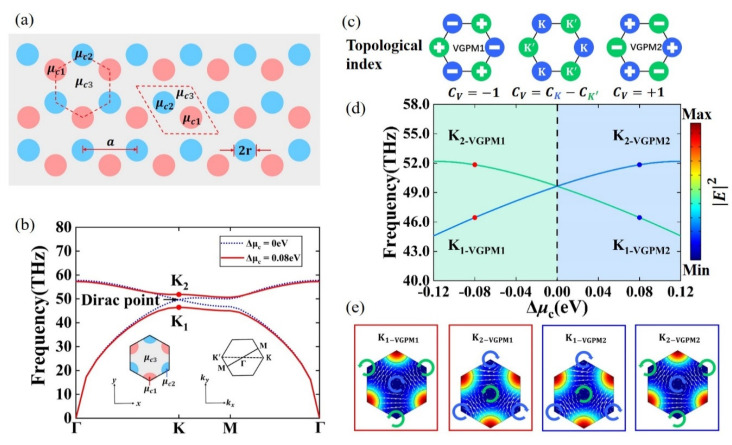
Band structure and valley Hall phase transition in VGPMs. (**a**) Schematic of VGPMs contains two types of unit cells (regular hexagon and rhombus), both of which are composed of two non-equivalent graphene nanodisks (scatterers) and ambient graphene (matrix). The chemical potentials of two graphene nanodisks and ambient graphene are $\mu_{c1}$, $\mu_{c2}$, and $\mu_{c3}$, respectively. The graphene nanodisks radii both are r=0.21a, where a is the lattice constant. (**b**) Bulk band for VGPMs with $\Delta \mu_{c}=\mu_{c1}-\mu_{c2}=0\;\mathrm{eV}$ and $\Delta \mu_{c}=0.08\;\mathrm{eV}$, respectively. Insets: hexagon unit cell (left) and the first Brillouin zone (BZ) (right). (**c**) Diagram of topological index distributions and theoretical valley Chern numbers for VGPM1 ($\Delta \mu_{c}<0\;\mathrm{eV}$) and VGPM2 ($\Delta \mu_{c}>0\;\mathrm{eV}$). (**d**) Phase diagram revealed by band-edge frequencies at the K_1_ and K_2_ valleys. Blue and green curves indicate the phase vortex corresponding to clockwise and anticlockwise power flows, respectively. Red/blue dots: frequencies for the designed VGPM1/VGPM2 at the K_1_/K_2_ valleys. (**e**) Photonic light fields and power flux field profiles at the K_1_ and K_2_ valleys corresponding to the red and blue dots in (**d**). Color: ${\left|E\right|}^{2}$; white arrows: power flux. Blue/green arc arrows represent typical vortex features.

**Figure 3 nanomaterials-11-01808-f003:**
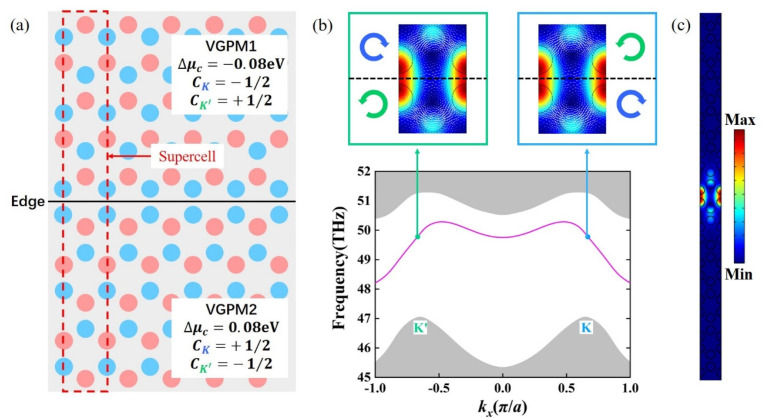
Valley topological edge states. (**a**) Schematic of the valley topological interface constructed by VGPM1 with $\Delta \mu_{c}=-0.08\;\mathrm{eV}$ (top) and VGPM2 with $\Delta \mu_{c}=0.08\;\mathrm{eV}$ (bottom). Red dotted rectangle: supercell for the valley topological interface. (**b**) Dispersion of the valley topological edge states (magenta line) for the designed supercell. Grey-shaded regions: projected TM polarized bulk bands. Insets: edge states at the K and K’ points. Color: ${\left|E\right|}^{2}$; white arrows: power flux. (**c**) The photonic light field for the designed supercell at the K/K’ points.

**Figure 4 nanomaterials-11-01808-f004:**
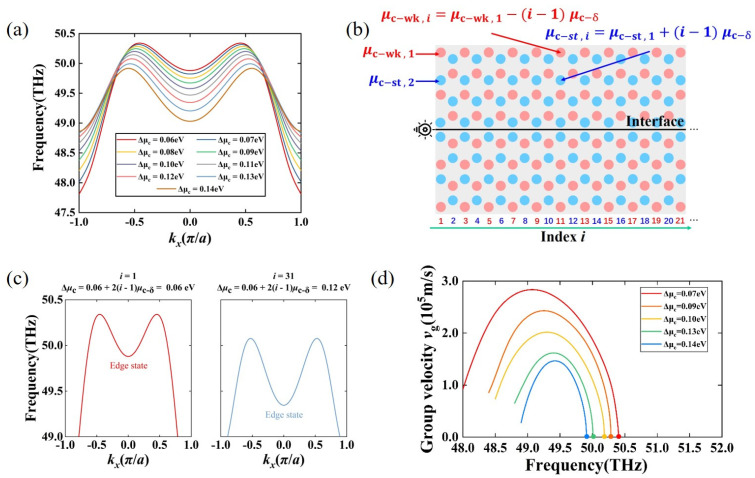
(**a**) Evolution of valley topological edge states with respect to the chemical potential difference $\Delta \mu_{c}$. With the increases of $\Delta \mu_{c}$, the edge state frequencies at the highest point gradually decreased. (**b**) Configuration of the chirped valley GPC edge state interface waveguide. Surface plasmon polariton (SPPs) wave along a straight interface exited on the left. The red and blue circles represent the graphene nanodisks of weak chemical potentials $\mu_{c-wk,i}$ and strong chemical potentials $\mu_{c-st,i}$. The weak and strong chemical potentials change as *i* increases as follows: $\mu_{c-wk,i}=\mu_{c-wk,1}-\left(i-1\right)\mu_{c-\delta }$ and $\mu_{c-st,i}=\mu_{c-st,1}+\left(i-1\right)\mu_{c-\delta }$. The graphene nanodisks of weak and strong chemical potentials in the first column were $\mu_{c-wk,1}=0.27\;\mathrm{eV}$ and $\mu_{c-st,1}=0.33\;\mathrm{eV}$, respectively. The step size $\mu_{c-\delta }$ was 0.005 eV. (**c**) Dispersion relation curves of the supercell with the chemical potential difference $\Delta \mu_{c}=0.06\;\mathrm{eV}$ in the 1st column and $\Delta \mu_{c}=0.12\;\mathrm{eV}$ in the 31st column. (**d**) Dispersion curves of valley edge state group velocities $v_{g}$ for the valley topological interface with $\Delta \mu_{c}$ from 0.07 eV to 0.14 eV.

**Figure 5 nanomaterials-11-01808-f005:**
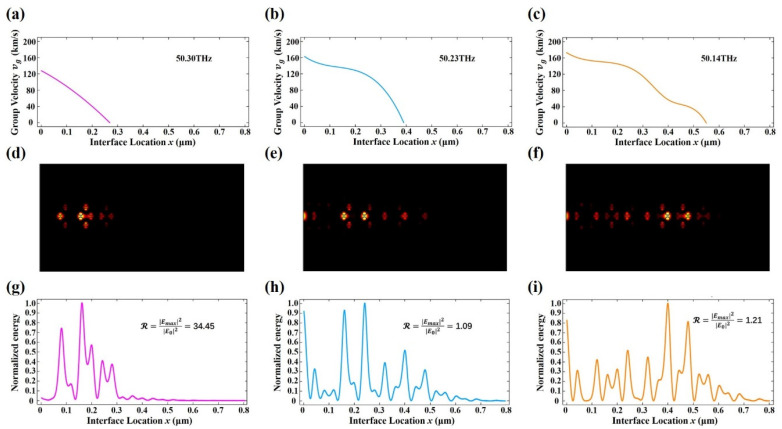
Variations of group velocities and light field of the edge SPP waves along with the interface of the chirped VGPM waveguide. (**a**–**c**) Calculated group velocities as a function of the interface location x at frequencies of 50.30, 50.23, and 50.14 THz. (**d**–**f**) Numerically experimental photonic light field fields of the edge SPP waves in the chirped VGPM waveguide at frequencies of 50.30, 50.23, and 50.14 THz. (**g**–**i**) Normalized edge SPP wave light field distributions corresponding to the simulated results along with the interface of the chirped VGPM waveguide. The intensity enhancement factors $ℛ=|E_{max}{|}^{2}/|E_{0}{|}^{2}$ were 34.45, 1.09, and 1.21, respectively.

## Data Availability

The data presented in this study are available on request from the corresponding author.

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
