# Peer review of "Group Velocity Modulation and Light Field Focusing of the Edge States in Chirped Valley Graphene Plasmonic Metamaterials"

_nanomaterials, 2021, doi:10.3390/nano11071808_

Round 1

Reviewer 1 Report

The paper entitled "Group Velocity Modulation and Light Field Focusing of the Edge States in Chirped Valley Graphene Plasmonic Metamaterials" by Zhuo et al. is a solid work, which is devoted to theoretical design of novel-type graphene-based plasmonic metamaterials with high promise in many applications, especially in optics and nanoplasmonics. By using COMSOL program package authors showed the principal possibility to tune the dispersion of valley topological edge mode through changing the chemical potential difference of valley graphene plasmonic metamaterials. The manuscript is well-organized, and well-written. Authors made significant contribution to the field of graphene plasmonics. The paper deserves immediate publishing in the current form.

Author Response

Dear Editors and Reviewers: 

Thank you for your letter and for the reviewers’ comments concerning our manuscript entitled “Group Velocity Modulation and Light Field Focusing of the Edge States in Chirped Valley Graphene Plasmonic Metamaterials”. These comments are all valuable and very helpful for revising and improving our paper, as well as the important guiding significance to our researchs. We have made revision according to the reviewer’s comments. The main corrections in the paper and the responds to the reviewer’s comments are as flowing:

Referee #1:

Special thanks to you for your good comments.

Reviewer 2 Report

The authors have reported theoretical proposals on new graphene plasmonic metamaterials. They propose nanostructures that consist of a monolayer graphene on a silica layer with a silicon substrate having cylinders with periodic thicknesses distributed in 2D honeycomb lattices. Local chemical potentials of the monolayer graphene can be controlled by this nanostructure. By changing the local chemical potentials gradually, they expect group velocity modulation and light field focusing of surface plasmon polariton waves. These results are useful for the development of new nanophotonic systems. However there are several points that should be revised as shown below.

  1. They have not reported the honeycomb lattice constant a. Is this about 100 nm? In Fig. 5 (that is wrongly shown as Fig.4), interface location x is written in a nanometer scale, which is too small for nanophotonics application. Is it a micrometer scale? They should show the actual honeycomb lattice constant, nanodisc radius, silica heights h and also confirm the scale of interface location.
  2. They have not reported how to make the metamaterial shown in Fig. 1. It may be difficult to form silica layer with flat surface on uneven Si substrate. It is also well-known that the silica layer affects the properties of the monolayer graphene. It is important to propose a real method to make the metamaterial when its application to the nanophotonic system is considered.
  3. In the references, there are many references that have wrong page numbers and no page numbers (References 2, 3, 7-9, 11, 13, 14, 17-19, 21-24, 27, 29, 36, 37,39 44-46). They should confirm the page numbers and revise them.

Author Response

Dear Editors and Reviewers: 

Thank you for your letter and for the reviewers’ comments concerning our manuscript entitled “Group Velocity Modulation and Light Field Focusing of the Edge States in Chirped Valley Graphene Plasmonic Metamaterials”. These comments are all valuable and very helpful for revising and improving our paper, as well as the important guiding significance to our researchs. We have made revision according to the reviewer’s comments. The main corrections in the paper and the responds to the reviewer’s comments are as flowing:

Referee #2:

Comment 1: They have not reported the honeycomb lattice constant a. Is this about 100 nm? In Fig. 5 (that is wrongly shown as Fig.4), interface location x is written in a nanometer scale, which is too small for nanophotonics application. Is it a micrometer scale? They should show the actual honeycomb lattice constant, nanodisc radius, silica heights h and also confirm the scale of interface location.

Response: Thanks for your serious review. We have shown the actual honeycomb lattice constant, nanodisc radius, silica heights h. The scale of interface location has also been corrected.

Comment 2: They have not reported how to make the metamaterial shown in Fig. 1. It may be difficult to form silica layer with flat surface on uneven Si substrate. It is also well-known that the silica layer affects the properties of the monolayer graphene. It is important to propose a real method to make the metamaterial when its application to the nanophotonic system is considered.

Response: Thanks for your serious review. Actually, the periodical pattern in nano-scale is definable with the electron-beam lithography technique. The electron dose is periodically modulated to get the periodical distribution of the height of the electron-beam resist. Then the pattern is transferred to the even Si substrate following by the planar growth of Silica on it to get the periodical distribution of the height of SiO2 with an even surface. Finally the graphene monolayer is transferred to the surface of the planar silica. On the other hand, this article mainly focuses on the design and analysis the group velocity of the valley state. The fabrication is expectable with the development of the advanced manufacture techniques.

Comment 3: In the references, there are many references that have wrong page numbers and no page numbers (References 2, 3, 7-9, 11, 13, 14, 17-19, 21-24, 27, 29, 36, 37,39 44-46). They should confirm the page numbers and revise them.

Response: Thanks for your serious review. We have confirmed and revised the page number of the references.

Thank you very much for your valuable advice. We appreciate for Editors/Reviewers’ warm work earnestly, and hope that the corrections will meet with approval. Once again, thank you very much for your comments and suggestions